# Comparing of Frequency Shift and Impedance Analysis Method Based on QCM Sensor for Measuring the Blood Viscosity

**DOI:** 10.3390/s22103804

**Published:** 2022-05-17

**Authors:** Shuang Liao, Peng Ye, Cheng Chen, Jie Zhang, Lin Xu, Feng Tan

**Affiliations:** School of Automation Engineering, University of Electronic Science and Technology of China, Chengdu 611731, China; liaoshuang1001@foxmail.com (S.L.); yepeng@uestc.edu.cn (P.Y.); chenchenguestc2020@163.com (C.C.); jiezhang6@163.com (J.Z.); xulin_zoe@163.com (L.X.)

**Keywords:** blood viscosity, quartz crystal microbalance (QCM), high precision, linear regression analysis, BVD circuit

## Abstract

Blood viscosity measurements are crucial for the diagnosis of cardiovascular and hematological diseases. Traditional blood viscosity measurements have obvious limitations because of their expensive equipment usage and large sample consumption. In this study, blood viscosity was measured by the oscillating circuit method and impedance analysis method based on single QCM. In addition, the effectiveness of two methods with high precision and less sample is proved by the experiments. Moreover, compared to the result from a standard rotational viscometer, the maximum relative errors of the proposed oscillating circuit method and impedance analysis method are ±5.2% and ±1.8%, respectively. A reliability test is performed by repeated measurement (N = 5), and the result shows that the standard deviation about 0.9% of impedance analysis is smaller than that of oscillating circuit method. Therefore, the impedance analysis method is superior. Further, the repeatability of impedance analysis method was evaluated by regression analysis method, and the correlation coefficient R^2^ > 0.965 demonstrated that it had excellent reproducibility.

## 1. Introduction

Blood viscosity is one of the important indexes reflecting blood flow and coagulation characteristics, which is an important condition for normal blood circulation. It plays an important role in the study of hemorheology. The change of blood viscosity will inevitably directly affect the blood flow velocity. For instance, the increased blood viscosity increases hematocrit and fibrinogen, which promotes ischemic heart disease, stroke and microvascular angina [1,2,3,4,5]; Elevated blood viscosity is associated with diabetes mellitus that may contribute to impaired tissue perfusion [6,7,8].

Traditional blood viscosity measurement mainly uses viscometers [9], including rotational and capillary viscometer, which are widely used in industry and laboratory research. The rotational viscometer employs a sensor to measure the shear stress that hinders the rotation of liquid, so as to obtain the liquid viscosity. On the other hand, Capillary viscometer is to force the liquid out of the capillary through the driving device for measuring the shear stress and shear speed, so as to obtain the liquid viscosity. However, although these conventional viscometers exhibit accurate and reliable measurement performance, these methods have obvious limitations in conventional use because of their expensive equipment and large sample consumption. These shortcomings increase the necessity of using microfluidic technology to measure and evaluate blood viscosity [10,11,12].

Recently, different techniques have been used in microfluidic devices to improve blood viscosity measurement. Microfluidic device [13] is composed of two inlets and a large number of indicating channels. The inlets are used to introduce sample and reference blood, respectively. Co-flowing device [14] is composed of a single inlet and two channels connected in parallel. Wheatstone device [15,16] consists of an inlet and a Wheatstone-bridge analog circuit. Based on the relationship between viscosity and shear rate characterized by two parameter power-law model, impedance analysis method uses nonlinear regression analysis to estimate blood viscosity. In addition, 3D printing capillary method [8], scanning capillary method [17] and molecularly imprinted method [18,19] are used for blood measurement. Blood viscosity measurement technology, which takes advantage of microfluidic technology, has the advantages of easy-of-use and small test samples, but it depends on the shear rate of blood.

Quartz crystal microbalance (QCM) with resolutions down to nanograms can make up for these deficiencies. The measurement system with several merits, such as low sample consumption, fast response, and high precision, can be adopted to measure blood viscosity. In addition, when a QCM sensor is applied in liquid phase, the detection principle is the well-known Kanazawa equation [20,21].
(1)$$\Delta f=-\frac{f_{0}^{3/2}\sqrt{\rho_{L}\eta_{L}}}{\sqrt{\pi \rho_{Q}\eta_{Q}}}$$
where $f_{0}$ is the resonance frequency, Δf is the frequency change due to the mass load, $\rho_{Q}$ is the density of quartz and $\eta_{Q}$ is the elastic constant of the piezoelectrically stiffened quartz, $\rho_{L}$ and $\eta_{L}$ denote the density and viscosity of the liquid, respectively.

According to the square root relationship between the QCM resonance frequency shift and the product of the liquid viscosity and density, Kanazawa equation can not only measure viscosity when the density is unknown. Up to now, numerous efforts have been paid to use QCM to measure liquid viscosity when the liquid density is unknown [22,23,24,25,26]. Dual QCM sensors, i.e., a smooth-surface resonator and a textured-surface resonator, which are used to measure viscosity is particularly complex, and without specific expression [27,28,29]. Based on admittance analysis, viscosity is measured using a single textured surface QCM (QCM-A) [30,31]. However, the measurement process is complex. In addition, expression depends on the surface texture of the sensor.

Our group [32,33,34] proposed oscillating circuit analysis method to measure liquid viscosity using single QCM with smooth electrode. Furthermore, this paper also used impedance analysis method to measure blood viscosity, and then compared the two methods.

The organizational structure of this paper is as follows: Section 2 describes the principle of blood viscosity measurement with QCM sensor based on oscillating circuit analysis method and impedance analysis method. Section 3 presents comparative analysis of the results for testing blood viscosity using two methods. The conclusions are drawn in Section 4.

## 2. Materials and Methods

### 2.1. Materials

The AT-cut 10-MHz QCM crystal with 5-mm smooth electrodes diameter on both sides is supplied by Chengdu Spaceon Electronics Co., Ltd. (Chengdu, China). Blood is mainly drawn from laboratory rabbits of Animal Laboratory College of Sichuan Agricultural University. Next, by adding distilled water to the blood, the blood concentration was diluted to 10% and 20%, respectively.

Experiments to measure blood viscosity with single QCM are conducted at room temperature (25 °C). There are two schematic diagrams of measuring the blood viscosity by QCM as shown in Figure 1. The solid line link is based on the oscillating circuit analysis method that is mainly consisted of QCM with thermostat, oscillator, a universal frequency counter (53132A, Agilent, Santa Clara, CA, USA), and TopPette (Dragon, Beijing, China) and a Laptop. The drive circuit is applied to oscillate QCM sensors. The frequency counter is utilized for real-time frequency acquisition, which is convenient to calculate the frequency shift. The dotted line link is based on impedance analysis method, which uses impedance analyzer (4294A, Agilent) to test.

### 2.2. Method 1: Oscillating Circuit Analysis Method

Oscillating circuit analysis method to measure the blood viscosity uses a QCM sensor, which is characterized by transforming the viscosity effect of sample blood into corresponding frequency shift.

Adding blood to the surface of the sensor by two drops ($V_{L1}$ and $V_{L2}$), theoretical expressions of blood viscosity $\eta_{L}$ is derived and shown as follows:(2)$$\eta_{L}=\frac{K_{P}C_{P}}{(\Delta f_{2}-\Delta f_{1})f_{0}\left(V_{L2}-V_{L1}\right)}{\left[\frac{\Delta f_{1}\left(V_{L2}-V_{L1}\right)-V_{L1}(\Delta f_{2}-\Delta f_{1})}{K_{T}C_{T}}\right]}^{2}$$
where $V_{L1}$ and $V_{L2}$ are the volume of blood added to the QCM at the first time and the second time, respectively; $\Delta f_{1}$ and $\Delta f_{2}$ are frequency shift caused by $V_{L1}$ and $V_{L2}$, respectively. $K_{P}$ and $K_{T}$ are the pressure sensitivity coefficient and the stress sensitivity coefficient of the QCM sensor, respectively.

According to the measurement principles discussed above, it is necessary to determine the pressure-frequency coefficient ($C_{P}$) and stress-frequency coefficient ($C_{T}$) in Equation (2). Pure water is chosen as the reference sample because the viscosity and density of that are known. Similarly, the blood viscosity can be measured by repeating the above experiment.

### 2.3. Method 2: Impedance Analysis Method

A modified equivalent Butterworth-van Dyke (BVD) circuit model of QCM without considering capacitance effect when it is coated with viscoelastic film, as shown in Figure 2, every electrical element in the BVD circuit reflects the different physical properties of crystal and viscoelastic film. L1, C1, R1, and C0 represent the dynamic inductance, dynamic capacitance, dynamic resistance and static capacitance of QCM, respectively; R2 and L2 describe the extra energy dissipation in the viscoelastic film and mass loading. Our group has deduced the relationship between R2 and load viscosity $\eta_{f}$ in previous work [34].
(3)$$\begin{array}{c} \Delta f=\frac{1}{2\pi \sqrt{\left\{\frac{8e_{26}^{2}l_{q}+\rho_{q}\omega^{2}{l_{q}}^{3}\varepsilon_{22}}{8\omega^{2}A_{s}\varepsilon_{22}e_{26}^{2}}+\frac{\rho_{f}l_{f}l_{q}}{4C_{0}K_{0}^{2}{\bar{c}}_{66}}\left[1+\frac{\omega^{2}}{3}\frac{{l_{f}}^{2}}{v_{f}^{2}}\right]\left[1+\frac{1}{4}{\left(\frac{\omega \eta_{q}}{{\bar{c}}_{66}}\right)}^{2}\right]\right\}\times \left\{\frac{8A{}_{s}e_{26}^{2}}{{\left(\pi \right)}^{2}l_{q}{\bar{c}}_{66}}\right\}}}\\ -\frac{1}{2\pi \sqrt{\frac{8e_{26}^{2}l_{q}+\rho_{q}\omega^{2}{l_{q}}^{3}\varepsilon_{22}}{8\omega^{2}A_{s}\varepsilon_{22}e_{26}^{2}}\times \frac{8A{}_{s}e_{26}^{2}}{{\left(\pi \right)}^{2}l_{q}{\bar{c}}_{66}}}} \end{array}$$
(4)$$L_{2}=\frac{\rho_{f}l_{f}l_{q}}{4C_{0}K_{0}^{2}v_{q}^{2}\rho_{q}}\left[1+\frac{\omega^{2}}{3}\frac{l_{f}{}^{2}}{v_{f}^{2}}\right]\left[1+\frac{1}{4}{\left(\frac{\omega \eta_{q}}{{\bar{c}}_{66}}\right)}^{2}\right]$$
(5)$$\eta_{f}=\frac{12C_{0}K_{0}^{2}v_{q}^{2}v_{f}^{4}\rho_{q}R_{2}}{\omega^{4}l_{f}{}^{3}l_{q}\left[1+\frac{1}{4}{\left(\frac{\omega \eta_{q}}{{\bar{c}}_{66}}\right)}^{2}\right]}$$
where ${\bar{c}}_{66}$, $e_{26}$ and $\varepsilon_{22}$ are the piezoelectric constants, elastic constants and dielectric constants of QCM, respectively. $K_{0}^{2}$ is the electromechanical coupling coefficient. $\rho_{q}$ and $\eta_{q}$ are the density and viscosity of QCM, respectively. The elastic wave velocities of $v_{q}$ and $v_{f}$ are QCM and blood, respectively. In this paper, AT-cut 10-MHz QCM is adopted. In addition, the characteristic parameters [35] are shown in Table 1.

By combining frequency Equation (3) with inductance Equation (4), the elastic wave velocity $v_{f}$ of QCM can be calculated, and then the blood viscosity can be calculated by Equation (5). Therefore, it is significantly necessary to monitor the change of R2 when blood drops on the QCM surface.

## 3. Results and Discussion

### 3.1. Oscillating Circuit Analysis Method

#### 3.1.1. Measurement of Frequency Coefficient

Due to the density and viscosity of pure water as known, the pressure frequency coefficient $C_{P}$ and stress frequency coefficient $C_{T}$ of QCM were calculated, as shown in Table 2.

#### 3.1.2. Measurement Blood Viscosity

During the experiments, the volume of the added blood was increased steadily 2.5 μL each time. The frequency shifts of QCM induced by different volumes were recorded as shown in Figure 3.

Figure 3 shows the dynamic frequency response of the QCM sensor based on oscillation circuit test method under the increase of blood volume. Noticeably, it can be observed that frequency decreases with increasing of blood volume. The sensor exhibits nonlinearity in volume less than 12.5 μL.The reason is that when the blood has not covered the whole sensor electrode (5-mm), the pressure effect and viscosity effect made the frequency of sensor shifted at the same time. Moreover, after the surface of the sensor is completely covered by blood (e.g., in the range of 12.5 μL–17.5 μL), the sensor exhibits an excellent linearity with blood volume, then only the pressure plays a crucial role.

After the electrode is covered completely, selecting any two volumes and its corresponding frequency shift, the blood viscosity can be calculated using Equation (2). In addition, the blood viscosity with concentrations of 10% and 20% is given in Table 3.

In Table 3, the reference value is measured with a standard rotational viscometer viscometer NDJ-5S (Shanghai Youyi Instrument Co., Ltd., Shanghai, China). Compared with five groups of experimental results and reference value, the values obtained by experiment are consistent with reference values. Furthermore, viscosity have an error about ±5.2% of the reference values. In addition, the standard deviation of the experimental results is 5.5%. It shows that it is effective to measure blood viscosity with only one QCM sensor by oscillating circuit analysis method.

### 3.2. Impedance Analysis Method

According to the BVD equivalent circuit, R2 represents the energy loss. The bigger the change in R2, the more energy lost it is. The resonance frequency, resistance and inductance of QCM before and after blood-droped were measured via impedance analyzer. L2 and R2 were obtained by subtracting the two values. Firstly, the elastic wave velocity was calculated, and then the blood viscosity was calculated by Equation (5). When the volume is 2.5 μL, the calculation results of blood viscosity is shown in Table 4.

According to the reference and measured viscosity, the effectiveness of BVD model in testing blood viscosity can be determined. In addition, viscosity have an error about ±1.8% of the reference viscosity. Moreover, the standard deviation of the experimental results is 0.9%. Therefore, it is obvious that this method has higher accuracy than the oscillating circuit method.

In addition, the repeatability experiment plays a vital role in evaluating the stability of blood viscosity by QCM sensor. As shown in Figure 4, the QCM sensor based on impedance analysis method was used to measure the blood with concentration of 10% and 20% for eight times, and the blood viscosity was calculated. Furthermore, the linear regression analysis method is used to analyze the eight groups of blood viscosity, and the coefficient of linear regression R^2^ > 0.965. Noticeably, the results confirmed that it had excellent reproducibility.

Experiments show that it is feasible to measure blood viscosity by QCM, which is based on oscillating circuit analysis and impedance analysis method. It is found that the measurement accuracy of impedance analysis is higher. In addition, compared with the references [11,36], the standard deviation of the experimental results in this paper is smaller. In the reference [11], the relative error and standard deviation of the experimental results are 5.3% and 2.19%, respectively. In the reference [36], the experiment used a scanning capillary tube viscometer to measure blood viscosity, the measurement standard deviation is 0.759.

The measurement error maybe caused by insufficient surface cleanliness of QCM sensor, because the sensor must be thoroughly cleaned and dried before each experiment; In addition, the blood viscosity is related to temperature, so inaccurate temperature control will also lead to errors.

## 4. Conclusions

In summary, the blood viscosity was measured by QCM based on oscillating circuit method and impedance analysis method. The experimental results demonstrate that the two methods possess high accuracy and low sample consumption properties in measuring blood viscosity. The test accuracy of oscillation circuit method is ±5.2%, and the blood volume is 17.5 μL; However, the accuracy of oscillation circuit method is ±1.8%, and the blood volume is 2.5 μL. In addition, the standard deviation about 0.9% of the test results of impedance analysis method is smaller than that about 5.5% of oscillation circuit method. It is found that the blood volume of impedance analysis method is less and the test accuracy is higher Therefore, the impedance analysis method is superior. Further, the repeatability of impedance analysis method is analyzed by regression analysis method, and the correlation coefficient R^2^ > 0.965, which demonstrated that it had excellent reproducibility. It provides a novel blood measurement method to evaluate the presence of anomalies in blood viscosity, which is an indicator of disease presence. On the basis of this paper, we are going to try our best to research the whole blood viscosity by QCM to make it a mature diagnostic technology in the future.

## Figures and Tables

**Figure 1 sensors-22-03804-f001:**
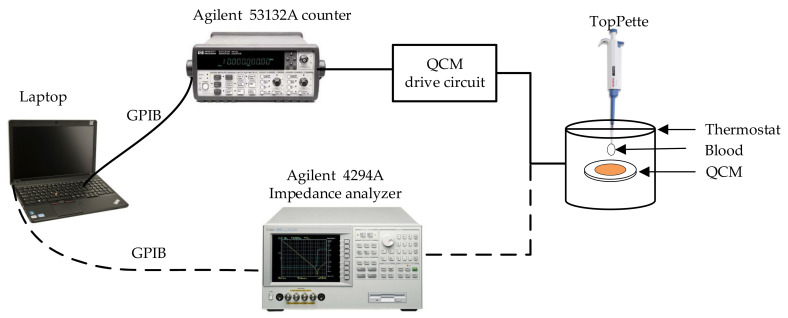
Test schematic for the measurement of viscosity by QCM.

**Figure 2 sensors-22-03804-f002:**
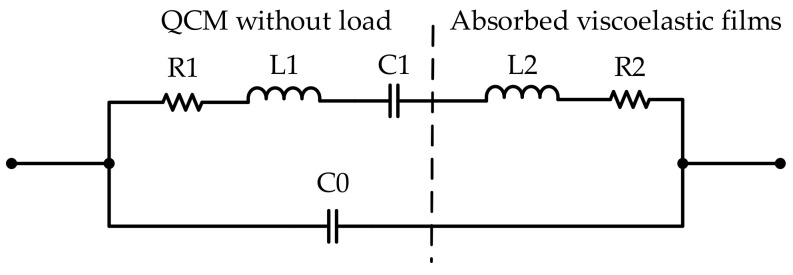
Equivalent BVD circuit with viscoelastic film loading.

**Figure 3 sensors-22-03804-f003:**
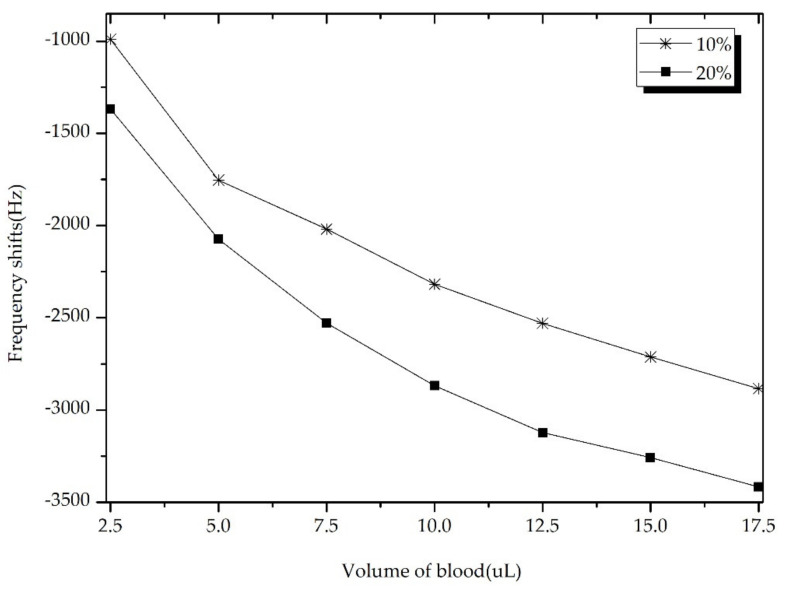
Response of the QCM sensor based on oscillating circuit analysis method.

**Figure 4 sensors-22-03804-f004:**
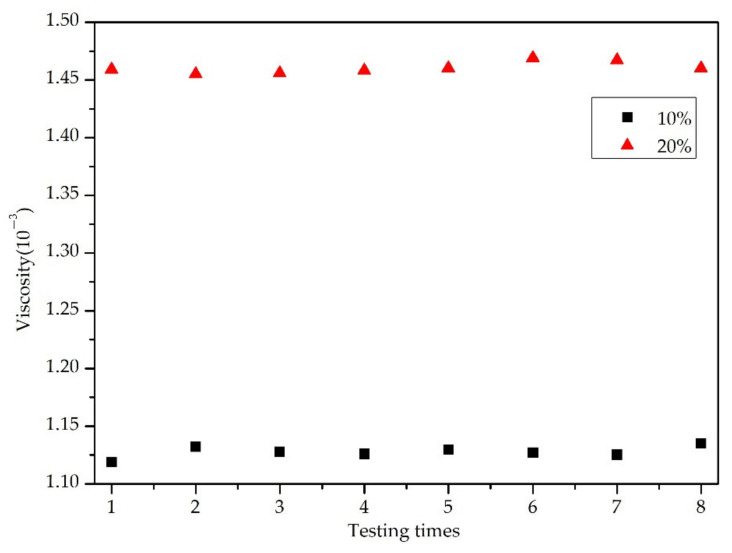
Repeatability experiment (N = 8) of blood viscosity QCM sensor based on impedance analysis method.

**Table 1 sensors-22-03804-t001:** Characteristic Parameters of QCM.

Parameters	Values	Units
$\rho_{q}$	2651	Kg·m^−3^
$\eta_{q}$	9.27 × 10^−3^	Pa·s
$A_{s}$	2.92 × 10^−5^	m^2^
$l_{q}$	1.66 × 10^−4^	m
$K_{0}^{2}$	7.44 × 10^−3^	
${\bar{c}}_{66}$	2.947 × 10^10^	N·m^−2^
$v_{q}$	3.32 × 10^3^	N^1/2^·Kg^−1/2^·m^5/2^
$e_{26}$	9.657 × 10^−2^	A·s·m^−2^
$\varepsilon_{22}$	3.982 × 10^−11^	A^2^·s^4^·Kg^−1^·m^−3^

**Table 2 sensors-22-03804-t002:** Measured pressure-frequency coefficient and stress-frequency coefficient.

$C_{P}$ $\;(m\times {\mathrm{Hz}}^{2})$	$C_{T}$ $\;(m\times \sqrt{{\mathrm{Hz}}^{3}})$
1.39 × 10^5^	−5.704 × 10^6^

**Table 3 sensors-22-03804-t003:** Measured the blood viscosity by oscillating circuit analysis method.

Concentration (wt%)	Viscosity (mPa·s)						
	Reference Value	Method 1 Measured Value					Standard Deviation
		1	2	3	4	5	
10	1.14	1.13	1.127	1.12	1.20	1.135	5.5%
20	1.48	1.44	1.455	1.456	1.458	1.41	2.1%

**Table 4 sensors-22-03804-t004:** Measured the blood viscosity by impedance analysis method.

Testing Times	Concentration (wt%)									
	10%					20%				
	R (Ω)	$v_{f}$ $\left(N^{1/2}\cdot {\mathrm{Kg}}^{-1/2}\cdot m^{5/2}\right)$	Measured Viscosity (mPa·s)	Reference Viscosity (mPa·s)	Standard Deviation	R (Ω)	$v_{f}$ $\left(N^{1/2}\cdot {\mathrm{Kg}}^{-1/2}\cdot m^{5/2}\right)$	Measured Viscosity (mPa·s)	Reference Viscosity (mPa·s)	Standard Deviation
1	83.56	1167.8	1.119	1.14	0.9%	102.05	1195.5	1.459	1.48	0.5%
2	84.03	1193.2	1.14			101.79	1189.9	1.465		
3	83.69	1186.7	1.12			101.9	1182.9	1.47		
4	83.72	1185.4	1.121			101.79	1200.1	1.458		
5	83.98	1189.1	1.13			101.59	1171.7	1.46		

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
