# Peer review of "Comparing of Frequency Shift and Impedance Analysis Method Based on QCM Sensor for Measuring the Blood Viscosity"

_sensors, 2022, doi:10.3390/s22103804_

Round 1

Reviewer 1 Report

Dear Editor,

The manuscript titled "Comparing of frequency shift and impedance analysis method based on QCM sensor for measuring the blood viscosity” reports blood viscosity measurements by oscillating circuit and impedance analyzer. The manuscript contains numerous typo and grammatical errors which need through correction.

This is an interesting study for measuring blood viscosity. I recommend a though major revisions of the manuscript before publishing in Sensors-mdpi. My specific comments are as follows.

  • In abstract authors stated that traditional blood viscosity measurement equipment is expensive. Please elaborate which equipment is expensive.
  • The abstract looks more general, it must include numerical values/figures for example, how much a sensor is sensitive and how many folds it is better, what are response/recovery times and etc.
  • Please cite some important literature concerning blood sensing for example,

Sensors 2016, 16(1), 51; https://doi.org/10.3390/s16010051

  • What is the novelty of current study?
  • What is the sensing layer?
  • Are blood samples diluted, if diluted, how much and what was the dilution medium? What do you mean by 10% and 20% concentrations?
  • Please specify the exact values of exposed surface area of QCM to blood samples?
  • How increasing the blood volume affects pressure?
  • The caption of figures 3 and 4 require more detailed description.
  • Only two values are compared with reference samples for measuring blood viscosity, why not more samples were measured?
  • Standard deviation is completely missing in the blood viscosity measurement data.
  • The mechanism of viscosity measurement by oscillating circuit and impedance analyzers are not thoroughly discussed.
  • What are the main advantages and limitations of current study, please describe in detail?
  • Moreover, the performance of current sensor should be compared with other potential strategies in tabular form explaining the key parameters.

Reviewer 2 Report

Review of the manuscript entitled ‘Comparing of frequency shift and impedance analysis method based on QCM sensor for measuring the blood viscosity’

The manuscript concerns the measurement of blood viscosity. Authors claims that the shear rate of blood must be measured with a new instrument and propose a QCM sensor.

Authors propose an oscillating circuit analysis method to measure liquid viscosity using single QCM with smooth electrode. The impedance analysis method is also used to measure blood viscosity, and then compare the two methods.

The work is very interesting and deserves to be published. However, I have some comments and remarks.

1) The manuscript contains some English spellings and grammatical mistakes that must be corrected.

2) When you do experiments with different blood volumes Are you measuring the blood density or the blood viscosity? I guess it is blood density more than blood viscosity. Could you explain? This is not clear how a static measurement could measure the viscosity of a fluid.

3) Have you evaluate the possible errors on the measurements with the application of propagating errors method by deriving the equation 3. How is the influence of the other parameter uncertainty on the final uncertainty of the frequency shift? So it is important to calculate the relative uncertainty.

4) A comparison with another method for the evaluation of the blood viscosity must be done.

5) First a viscosity is link to a flow. A QCM sensor is used to measure a mass. How do link viscosity and mass?

Reviewer 3 Report

I do not have any important comments to the work entitled "Comparing of frequency shift and impedance analysis method based on QCM sensor for measuring the blood viscosity". I would like just to ask Authors to check the manuscript from editing point of view - there are some inaccuracies.

Round 2

Reviewer 1 Report

Most of my comments are adequately addressed however, the performance of current sensor is not compared with other potential strategies. 

Reviewer 2 Report

The authors have correctly responds to the reviewer's remarks and comments. The manuscript can now be accepted.

Author Response

We appreciate the your comments that have helped us to improve the manuscript.